# Differential Anti-Inflammatory Effects of Electrostimulation in a Standardized Setting

**DOI:** 10.3390/ijms25189808

**Published:** 2024-09-11

**Authors:** Biagio Di Pietro, Simona Villata, Simeone Dal Monego, Margherita Degasperi, Veronica Ghini, Tiziana Guarnieri, Anna Plaksienko, Yuanhua Liu, Valentina Pecchioli, Luigi Manni, Leonardo Tenori, Danilo Licastro, Claudia Angelini, Lucia Napione, Francesca Frascella, Christine Nardini

**Affiliations:** 1Consiglio Nazionale delle Ricerche, Istituto per le Applicazioni del Calcolo “Mauro Picone”, 00185 Roma, Italy; dipietro@iac.cnr.it (B.D.P.); tiziana.guarnieri@unibo.it (T.G.); anna.plaksienko@medisin.uio.no (A.P.); liuyuanhua@tongji.edu.cn (Y.L.); claudia.angelini@cnr.it (C.A.); 2Dipartimento di Scienza Applicata e Tecnologia, Politecnico di Torino, 10129 Turin, Italy; simona.villata@polito.it (S.V.); francesca.frascella@polito.it (F.F.); 3PolitoBIOMed Lab, Politecnico di Torino, 10129 Turin, Italy; 4Area Science Park, Basovizza, 34149 Trieste, Italy; simeone.dalmonego@areasciencepark.it (S.D.M.); margherita.degasperi@areasciencepark.it (M.D.); danilo.licastro@areasciencepark.it (D.L.); 5Department of Chemistry “Ugo Schiff” and Magnetic Resonance Center (CERM), University of Florence, 50019 Sesto Fiorentino, Italy; ghini@cerm.unifi.it (V.G.); tenori@cerm.unifi.it (L.T.); 6Dipartimento di Scienze Biologiche, Geologiche e Ambientali (BIGEA), University of Bologna, 40100 Bologna, Italy; 7Oslo Center of Biostatistics and Epidemiology, University of Oslo, 0317 Oslo, Norway; 8State Key Laboratory of Cardiology, Shanghai East Hospital, Tongji University School of Medicine, Shanghai 200120, China; 9Consorzio Interuniversitario Risonanze Magnetiche Metallo Proteine (CIRMMP), 50019 Sesto Fiorentino, Italy; valentina.pecchioli@edu.unifi.it; 10Consiglio Nazionale delle Ricerche, Istituto di Farmacologia Traslazionale, 00185 Roma, Italy; luigi.manni@ift.cnr.it

**Keywords:** inflammation, electrostimulation, wound healing, 3D bioconstruct, fibroblasts, keratinocytes, omics

## Abstract

The therapeutic usage of physical stimuli is framed in a highly heterogeneous research area, with variable levels of maturity and of translatability into clinical application. In particular, electrostimulation is deeply studied for its application on the autonomous nervous system, but less is known about the anti- inflammatory effects of such stimuli beyond the *inflammatory reflex*. Further, reproducibility and meta-analyses are extremely challenging, owing to the limited rationale on dosage and experimental standardization. It is specifically to address the fundamental question on the anti-inflammatory effects of electricity on biological systems, that we propose a series of controlled experiments on the effects of direct and alternate current delivered on a standardized 3D bioconstruct constituted by fibroblasts and keratinocytes in a collagen matrix, in the presence or absence of TNF-α as conventional inflammation inducer. This selected but systematic exploration, with transcriptomics backed by metabolomics at specific time points allows to obtain the first systemic overview of the biological functions at stake, highlighting the differential anti-inflammatory potential of such approaches, with promising results for 5 V direct current stimuli, correlating with the wound healing process. With our results, we wish to set the base for a rigorous systematic approach to the problem, fundamental towards future elucidations of the detailed mechanisms at stake, highlighting both the healing and damaging potential of such approaches.

## 1. Introduction

The therapeutic usage of physical stimuli is a highly heterogeneous research area [1]. Such heterogeneity derives from stimulus-specific availability of biological knowledge, fragmentation of the research area and lack of standardization -and hence limited reproducibility- that overall limit translation to very early stages (experimental therapies) or to complementary and alternative medical approaches (CAM). We recently discussed how these features echo also in the *medical discourse* [2] and how the importance of having a unified view on the therapeutic effects of physical stimuli could be, instead, carefully explored for its potential impact on inflammation [3].

Following up on these motivating works, we here present our first steps to address the above limitations, i.e., the production of a 3D in vitro model including human fibroblasts embedded in a collagen matrix with human keratinocytes seeded on top [4], an approach that facilitates standardization. In this context, we explore the controlled and reproducible effects of physical stimuli on inflammation, if any. Electrotherapy takes a variety of names, rationales and protocols. The most recent and solid results refer to effects on the autonomic nervous system [5], whose full translation to medicine is still under exploration (see for instance medical application in rheumatoid arthritis [6] and Alzheimer disease [7]). Well known and controversial are also electrostimulations released in the frame of traditional medicines, that, despite very similar protocols and recent advances in the field of neurophysiology [8,9], rarely share the same dissemination, medical and health policy channels.

Our hypothesis is that the underlying function and overall conceptual umbrella to explain a large part of the therapeutic effects of physical stimuli, and hence of electrostimulation, lies in the *wound healing* function (WH, including at the cellular and molecular scale epithelial-mesenchymal transition Type II, EMT II [10]). It is according to this hypothesis [3], our early results on animal models [11] and human pilot study [1], that we chose to apply electrostimulation to a purposefully simply engineered substrate [4]. In fact, this set up is deprived of nervous, vascular and immune system, but equipped with realistic 3D features (fundamental in immune and regenerating activities typical of *WH* [12]) to assess whether very basic, highly conserved induction of EMT type II could be observed, under which circumstances and with what effect. *WH* proceeds from an early (one to few hours) activation of *inflammation* towards *proliferation* and *regeneration* observed for weeks and months in complex organisms, and already around 48 h in simplified models, a fact exploited in our experimental setting (more information can be found in Appendix A).

Our experiment involves the testing of four stimuli: two in direct current at 1 V and 5 V, and two in alternate current at 5 V with a frequency of 10 Hz and 100 Hz (labels 1DC, 5DC, 10AC, 100AC, respectively and NO for the absence of stimuli) applied to physiological (3D in vitro model, label PHYS) and inflamed (same bioconstruct perfused with TNF-α, label INLF) *states*, assessed at three meaningful time points (baseline 0 h, 1 h for the potential identification of early genes and 48 h for steady state effects, labels: 0, 1, 48, respectively), see further details in Methods and Appendix A. Of note, TNF-α administration used to model the inflamed state is a simplified and effective *ad hoc* method as previously reported [13,14,15], based on the crucial activity that TNF-α exerts as major pro-inflammatory agent. Given the complexity of the inflammatory condition, the TNF-α -induced inflamed state model is considered a good approximation of the pathological state, without claiming to be an exhaustive representation of the extremely intricate inflammatory circumstances observed in vivo.

The analysis via transcriptomics and metabolomics of the contrasts (differential analysis) shows that, under the explored circumstances, AC effects do not seem to be directly relevant in terms of the inflammatory state, but have an impact on proliferation that is state- and frequency-dependent; that the inflammatory state can be resolved with a 5VDC stimulus; and that 1VDC on physiological samples is able to elicit a durable inflammatory state. To the best of our knowledge this is the first systematic experimental exploration of this type, complementing a recent and very complete review by Katoh [16] covering the effect on wound healing of a variety of electrostimuli on different cell types and organisms. However, in that work, the large collection of heterogeneous experiments did not allow the systematic comparison of differential electrical effects, one of the main lacks in this research area. Our work, coping exactly with these limitations, leads to original and yet unknown insights into the impact and potential of electrostimulation, highlighting for the first time, how very different biological function can be elicited under the same conditions, with different electrical parameters. While this only scratches the surface of the potential and limitations of electrostimuli in an anti-inflammatory key, it also provides a framework, the first, for the systematic and systemic observation of such different phenomena, key for future causal exploration. Further investigations to complete this initial systematic exploration is crucial to proper clinical translation.

## 2. Results and Discussion

To approach the large datasets obtained with our experimental design (Table 1), we broke our analysis in the investigation of six questions (left column of Table 2) built to progressively capture complexity. Each question is addressed via enrichment analysis of differentially expressed genes (DEGs), obtained from all contrasts deemed relevant and listed in the second column of Table 2. The semantics of our DEGs is explored via the Hallmark gene sets [17]. The third column of Table 2 includes a summary of the effects observed, as they will be discussed in more details below.

For each of the six questions, results are displayed in large figures with, first, a bird-eye view (top heatmap) on the structure of the enrichment by *Groups*, composed of manually-curated selections of Hallmark gene sets, based on their biological functions and described in Table 3. Color intensity represents the normalized count of significant Hallmarks, described in details in Section 3.4 with darker shades indicating higher counts and higher enrichment, irrespective of the up- or down-regulation, favoring the inclusive identification of meaningful patterns. It is pure coincidence that Questions and Groups both appear in the number of six.

The biological insight obtained with the heatmap is then explored at the Hallmarks level, by means of an enrichment dot plot, enabling the assessment of the significance and direction (up- down- regulation) of the enrichment, facilitating biological considerations. Note that only Hallmarks that present a significant enrichment are represented, i.e., rarely the dotplot includes all fifty *Hallmark* functions.

It is worth remarking here once more the terminology adopted: we indicate as INFL the samples representing a chronic inflammatory condition, with PHYS or INFL representing the state of the sample. Group 2 is concerned with the enrichment for *Immune Response and Inflammation*, which in turn can indicate the enrichment for a number of the included *Hallmark* inflammatory functions/activity listed in Table 3.

### 2.1. Question 1: Impact of Time

From the heatmaps in Figure 1 we observe that, as expected, there is limited variation over time in the PHYS samples, and in particular Group 2 (associated with inflammation) shows mild to no enrichment. Similarly, when comparing PHYS versus INFL, we observe the desired enrichment in Group 2 (by experimental design, via TNF-α perfusion). Also, we observe for the INFL samples, variations for the enrichment in Group 2, indicating that the inflammatory state undergoes some variation over time.

By zooming on the dotplot in Figure 1, focusing in particular on the functions involved within Group 2, we observe that, although INFL samples at 48 h show a reduction in the inflammatory activity compared with earlier time points, comparison with the PHYS state at homologous time points returns continued overexpression of the inflammatory functions, in particular with reference to *TNF-α signaling via NF-kB* that remains stable, confirming the appropriateness of the inflammatory experimental model we chose, despite spontaneous exhaustion of other concomitant inflammatory functions with in particular *IL6 JAK/STAT3 signaling*, *complement* and *allograft rejection* undergoing visible reduction in activity.

Continuing with the exploration of Group 3, INFL is accompanied by a decrease of *E2F targets*, *G2M checkpoint*, *MYC target 1-2* and *mitotic spindle* activities. These functions reduce their activity over time also in the PHYS samples, however, divergence between INFL and PHYS at homologous time-points increases over time i.e., while inflammation remains higher, the activities connected with these functions (Group 3 cellular regulation and proliferation) decrease more significantly in INFL. These functions are all related to the regulation of the cell cycle, cell division and gene expression, and play crucial roles in ensuring proper cell growth and proliferation. In a physiologic environment, this effect is normal, due to the slowly ageing of the 3D structure; an inflammatory environment can inhibit cell cycle progression, thus affecting the expression of these genes. Shall this phenomenon of reduced proliferation chronicise, it could be associated to cellular senescence [18]. This is also accompanied by a higher level of *hypoxia*, slowly increasing over time in PHYS samples, but nevertheless lower than the stable level observed in INFL.

This information is backed by one of the few significant negative enrichment from the metabolomics analysis relative to the Kyoto Encyclopedia of Genes and Genomes (KEGG) *Valine, leucine and isoleucine biosynthesis* pathway in four of the nine contrasts of this Question (PHYS.48vs0.NO, INFL.1vs0.NO, INFL.48vs1.NO and INFL.48vs0.NO, Appendix A), indicating that we observe a decrease in the uptake in the corresponding conditions. These metabolites are collectively known as *branched chain amminoacids* (BCAA) and are fundamental in numerous functions (lipid and glucose metabolism, regulation of the immune response, to name a few [19]). The four enriched contrasts belong, two by two, to the purple (low activity) and yellow (inflammation) cluster, respectively, of the top heatmap in Figure 1 indicating that the reduced uptake of such metabolites can lead to different inflammatory effects (yellow versus purple activity). However, these contrasts do share a common transcriptional activity i.e., a reduction in *Epithelial mesenchymal transition (EMT)* (Group 4). In this case, the BCAA reduced availability seems to correspond to a reduction in genesis and proliferation. These data could be linked to a time-dependent decrease in nutrients in the culture media of both physiological and inflamed specimens. The consequent hypoxia could have instigated an endoplasmic reticulum stress [20], which, together with inflammation is able to regulate BCAA uptake and metabolism, as highlighted by Burrill and colleagues [21]. At this stage of our understanding, however, this remains reasonable speculation.

In summary, we observe for both *states* of the samples (INFL and PHYS) a reduction of the proliferative activity, possibly attributable to the physiological cellular senescence of the sample, faster in INFL, with PHYS accompanied by lower but increasing hypoxia, and INFL by a mild reduction of inflammation, possibly due to compensation mechanisms, as the inflammatory stimulus was administered once before baseline. Importantly, these variations are dominated by a more striking difference of the inflammatory machinery, whose activity is predominant in the INFL samples.

### 2.2. Question 2: Impact of Stimuli on Physiological Samples

In the second heatmap (Figure 2) we observe that in PHYS, although DC1, 5 and AC10 are able to transiently (1 h) modify the inflammatory activity, only DC1 and AC100 can elicit a stable (48 h) inflammatory state (yellow cluster, Group 2).

A closer look in Figure 2 at Group 2 functions highlights that AC100 and DC1 elicit opposite effects, i.e., a mild reduction and a marked activation, respectively, of interferons’ activity. A more general observation, beyond Group 2, (i.e., all Hallmarks Groups), shows as the result of the AC100 stimulus, in comparison with the PHYS baseline, a reduction in: *Interferonγ-α* activities, *oxidative phosphorylation* and *fatty acid metabolism*. Interferons are known to be involved in numerous functions, beyond the adaptive immune response, which include also cell differentiation, cell growth and overall anti-tumor functions [22].

Here also, this information is backed by the significant positive enrichment from the metabolomics analysis relative to the *Valine, leucine and isoleucine biosynthesis* pathway (PHYS. 48. AC100vsNO, Appendix A), indicating that the cells of our sample show an increase in uptake of such metabolites 48 h after the AC100 stimulus. *BCAA* are also fundamental in the regulation of *lipolysis* and of the innate immune response [19], corroborating the findings of the transcriptional enrichment.

In summary, AC100 synergizes the pro-inflammatory effects with the downregulation of a factor (IFN-γ), known to be involved in numerous functions which encompass cell differentiation, cell growth and anti-tumor functions, suggesting the potential harmfulness of this type of stimulation on non-inflamed tissues and systems. Overall this may suggest that while direct current seems to fuel inflammation and even *transitional* activity (i.e., referred to the process associated with a transition from a state to another) in normal specimens, alternate current seems to depress cell metabolism and, consequently, cell renewal.

Other stimuli in the long term (48 h) are able to provoke different effects, including: the deactivation of *EMT* and upregulation of *MYC*, generally associated with tumorigenic phenomena [23]. For DC5 and AC10 a pattern similar to the cellular senescence observed in Section 2.1, i.e., the deactivation of proliferative functions within Cluster 3 can also be detected.

### 2.3. Question 3: Effects of Stimuli on Inflamed Samples

Interestingly, all stimuli appear to elicit some transient (1 h) inflammatory activity (Group 2) that is, however, exhausted at 48 h for all but the DC5 stimulus, presenting a striking and peculiar patterning.

When zooming at the Hallmark level—Figure 3—on Group 2 the pattern is even more clear, with the TNF-α activity being the only transiently (1 h) exacerbated for all stimuli, and only DC5 being affected at 48 h (downregulated), a pattern in line with *WH* progression. This is confirmed in the overall enrichment picture, where DC5 at 48 h is the only contrast enriched for *EMT*. At the same time, Group 3 functions appear to be upregulated, indicating enhanced proliferation, part of the expected activity of the second phase of *WH*.

Another generally downregulated late (48 h) profile is offered by AC100, where, however, the activity is focused on the downregulation of *cell cycle* and *energy metabolism* (Groups 3 and 6, respectively).

### 2.4. Question 4: Impact of States (INFL and PHYS) on Stimulus Effects

Two patterns are clearly highlighted in the heatmap in Figure 4. One (purple) related to the absence of enrichment for Group 2, i.e., absence of differential inflammation in the two INFL and PHYS *states*, and one (yellow) grouping contrasts that present a very broad range of variations across all Groups.

Regarding the first cluster, given the initial *states*, the absence of enrichment has to be attributed to an action—by both ACs—that cancels out the inflammatory differences existing between the *states*, which could include: (i) inflammation of PHYS samples, (ii) complete resolution of inflammation of INFL or (iii) a mild anti-inflammatory action on INFL and an inflammatory action on PHYS, Questions 5 and 6 will shed light on these points. This inflammatory effect is also accompanied (as already seen above) by apparently opposite effects, with AC100 eliciting a reduction and AC10 an increase on *MYC target V1-2*, *E2F target*, *G2M Checkpoint* (Group 3). Since *MYC target V1*, *MYC target V2*, *E2F target*, and *G2M Checkpoint* are markers related to different phases of the cell cycle and cell proliferation regulation, AC100 and AC10 appear to exert an opposite effect on cell cycle progression. Additionally AC10 is accompanied by the reduction of *Bile acids* (linked to cholesterol) and *phospholipid* metabolisms, possibly involving the structure and the physiology of the cell membranes, while AC100 is accompanied by a reduction of *oxidative phosphorylation*, indicating a reduction of the aerobic production of energy, a marker of metabolism depression.

Regarding the second cluster (yellow), the overall dotplot in Figure 4 gives more insight into the late activity (48 h) of DC stimuli, characterized by very different patterns of activation, both, however, generally downregulated with the exception of proliferative functions (Group 3). Inspection of the direct comparison in Questions 6 (INFL.48 DC1vsDC5) shows a more significant anti-inflammatory action of DC5 over DC1 in INFL. The general dotplot highlights the modification of *EMT*, down at 1 h in DC1, and up at 48 h in DC5. This may explain the less decisive anti-inflammatory action of DC1, for which, we would otherwise expect an early (1 h) upregulation of inflammation to represent an effective *WH* process (observed indeed in DC5).

### 2.5. Question 5: Differential Impact of Stimuli on PHYS

As above (i.e. Figure 4) two main patterns are also visible in Figure 5, the first (yellow) with a general activation of all Groups, and the second (purple) with milder activity. At this stage several of the observations portrayed above can be confirmed and further clarified. In particular the long term (48 h) effect of DC5 and ACs leading to the absence of differential expression can be the results of ineffective stimuli or equally effective stimuli. Based on the observations above, we conclude that a lasting (48 h) and marked differential inflammatory activity (Group 2) is confirmed by DC1 versus all stimuli with a seemingly pro-inflammatory activation of DC1 (versus baseline), while other stimuli affect only proliferative functions, as discussed already in Question 2.

Over time, PHYS samples stimulated with DC1, versus AC, appear to move towards a maturation of the pro-inflammatory response, i.e., from a simpler increase of the TNF-α mediated response, towards the activation of interferons.

### 2.6. Question 6: Differential Impact of Stimuli on INFL

The heatmap in Figure 6 highlights a situation where some of the considerations made on the PHYS samples above (Question 5) seem to hold by directly replacing DC5 with DC1.

In particular in INFL, DC1 and AC10 do not show significant lasting (48 h) enrichment, indicating the same type of effect, and DC1 seems to stimulate an activity that is more proliferative than AC100, and, in turn, AC100 less proliferative than AC10. Finally, AC100 samples show reduced cell cycle activity and energy production and DC5 stronger anti-inflammatory and proliferative activity when compared to AC100 and to baseline.

## 3. Materials and Methods

### 3.1. Study Design

To obtain the 3D in vitro model, for each sample 350 μL of type 1 rat-tail collagen (Roche, Basel, Switzerland) dissolved in acetic acid 0.2% v/v was supplemented with 15 μL of sterile $ddH_{2}O$, 35 μL of NaOH 0.5 M, 50 μL HEPES 0.2 M, 50 μL DMEM 10X (Sigma Aldrich, St. Louis, MO, USA) and 25 μL of human fibroblasts (HFF-1, ATCC) at a final concentration of $1.5\times 10^{6}$ cells/ml of mixture. 500 μL of HFF-1-laden collagen was seeded onto PET 12-well hanging inserts with a porosity of 0.4 μm (Milicell^®^, Merck, Rahway, NJ, USA) and let gelify for 30 min at 37 °C. Once the stable dermal layer was established, human keratinocytes (HaCaT, Antibody Research Corporation, St. Charles, MO, USA) were seeded on top of the dermis at a density of $2\times 10^{5}$ cells/insert. The first three days of culture were conducted in a submerged environment using complete DMEM (15% FBS, 2% L-glutamine, 1% penicillin-streptomycin, 1% sodium pyruvate) to facilitate keratinocyte adhesion to the dermis. After three days, Air-Liquid Interface (ALI) culture was initiated by providing 400 μL of 3dGRO^TM^ Skin Differentiation Medium (Sigma Aldrich, St. Louis, MO, USA) in the lower compartment while leaving the upper compartment exposed to air. The culture was maintained for a total of fourteen days.

To mimic the inflamed state, TNF-α (Abcam ab259410, Cambridge, UK) was administered at a concentration of 50 ng/mL in the culture medium beneath the skin model (total volume 400 μL) the day before electrostimulation. After 24 h (just before electrostimulation) and the subsequent day (24 h post-stimulation), 200 μL of fresh medium were added to ensure metabolically active cells, without further addition of TNF-α [13,14,15]. Electrostimulation was released (only once, at baseline) on the constructs by two stainless steel sterile acupuncture needles with a diameter of 200 μm and a distance between them of 10 mm via *TENS* machine electrostimulator QiuTian Model SDZ II (Acquaviva, Repubblica di San Marino) and Hewlett Packard (Palo Alto, California, USA) Power Supply for AC and DC stimuli, respectively for 20 s, as described in [4].

Beyond baseline, the first sampling time point was set at 1 h after the electrostimulation as in [11] to assess the activity of early genes, and at 48 h as a reasonable compromise to guarantee the stability of the bioconstruct and the relevance of the stimulus under study (see Appendix A for this specific choice). All samples were run in triplicates, except for the non-stimulated physiological and inflamed samples at time 1 h, in duplicates.

The overall study design is shown in Table 1, additional information on the rationale of the study can be found in Appendix A.

### 3.2. Omics

#### 3.2.1. Transcriptomics

RNA was extracted using Quick-DNA/RNA™ Microprep Plus Kit from Zymo Research (Irvine, CA, USA and stored at −80 °C before paired-end 150 bp length sequencing (2 × 150 bp) was run on a Illumina (San Diego, CA, USA) *Novaseq 6000* machine for all conditions and time points (data accessible at NCBI GEO database, ID *GSE254696*).

#### 3.2.2. Metabolomics

NMR-based metabolomic analysis was performed on cell growth media according to standard procedures [24]. H1 NMR spectra were recorded with a Bruker 600 MHz spectrometer (Bruker BioSpin, Billerica, Massachusetts, Stati Uniti) optimized for metabolomic analysis, equipped with a 5 mm PATXI H1C˘13N˘15 and H2-decoupling probe. All samples were acquired with the Carr–Purcell– Meiboom–Gill (CPMG) at 310 K using 512 scans, 73,728 data points, a spectral width of 12,019 Hz and a relaxation delay of 4 s. The raw data were multiplied by a 0.3 Hz exponential line broadening before applying Fourier transform.

The metabolite assignment was performed using Chenomx software V9. The relative quantification of the NMR signals was performed using a R script developed in-house. The metabolites’ abundance levels for each contrast are listed in Appendix A. Twenty five metabolites were quantified and divided into two different groups: (i) molecules *taken-up* from the growth medium during cellular growth; (ii) molecules *released* into the medium during cellular growth, as shown in Appendix A.

### 3.3. Differential Analysis

#### 3.3.1. Transcriptomics

Analysis for the identification of DEGs was run with *DeSeq2* R package [25], testing the null hypothesis that the groups identified based on the parameters in Table 1 have the same gene expression, using a regression model with three predictors (State, Time and Stimulus), results are listed in Appendix A.

#### 3.3.2. Metabolomics

Data were pre-processed by removing the rows for the *blank* signals. After *logit* transformation of the metabolite concentrations, the same design matrix used for transcriptomics was used to model the interactions between three predictors (State, Time and Stimulus). The linear model was fitted to the data, and empirical Bayes moderation was applied to obtain more stable estimates of the log-fold changes and their associated statistics, using the R *limma* package [26,27]. The ${\mathrm{Log}}_{2}FoldChange\left(FC\right)$ was calculated to display how metabolite levels vary between groups. For the *released* molecules, ${\mathrm{Log}}_{2}FC>0$ indicates increased release; conversely, for the molecules *uptaken* from the growth media, ${\mathrm{Log}}_{2}FC>0$ indicates reduced consumption.

### 3.4. Enrichment

To gain insight into the biological functions at stake in our experiments, enrichment was first run on transcriptomics, given the dominant throughput of this omic (≈15,000 transcripts versus 25 metabolites).

#### 3.4.1. Transcriptomics

*Hallmarks Gene sets Enrichment* Gene set enrichment analysis [28] was used to assess the biological semantics of the molecules identified from the differential analysis ranked based on the Wald statistic $log_{2}\left(FC\right)/lfcSE$, where lfcSE is the log-fold change Standard Error, calculated with DESeq, using the function *GSEA* in the R package *clusterProfiler* [29]. The function returns a normalized enrichment score (*NES*), the corresponding statistical significance (*q*-value, computed by permutation) and the *core* list of genes contributing to the *NES* score, as shown in Appendix A.

*Hallmarks Groups Enrichment* To facilitate the initial readout, given the numerous contrasts and gene sets, we manually curated the grouping of the 50 Hallmarks into six more general functional *Groups* described in Table 3. The magnitude of the enrichment for each Group is computed as the number of statistically significant Hallmarks set composing the Group (adjusted *q*-value < 0.01), divided by the number of Hallmarks sets in the Group. Therefore, the *adjusted q*-value is the *q*-value (implicitly corrected for the number of gene sets, i.e., 50 Hallmarks), further Bonferroni-corrected by the number of contrasts. A clustered heatmap (R function *ComplexHeatmap* [30] (agglomeration method *‘ward.D2’* and Euclidean distances) is used to facilitate the readout as shown in Section 2.

#### 3.4.2. Metabolomics KEGG Enrichment

To assess the contribution of metabolites to the enrichment analysis, GSEA was performed on the combined transcriptomic and metabolomic data against KEGG [31] given its specific focus on metabolic functions. We first converted metabolites in Human Metabolome Database (HMDB) IDs and transcripts in Ensembl IDs [32,33]. We then ranked both types of features based on the *local statistics* (ls, function *rankFeatures* from R package *fgsea* [34]) as $ls=log_{2}\left(FC\right)/log_{10}$ (*p*-value) with *FCs* and *p*-values returned by *DEseq* and *limma*, for transcripts and metabolites, respectively (Appendix A). This joint (heterogeneous) KEGG enrichment was then compared to the transcriptomics-only KEGG enrichment (Appendix A) to identify the functions enriched only in the joint analysis, thus indicating a specific contribution of metabolomics, if any (Appendix A).

## 4. Conclusions

The overall experimental design allows to highlight several interesting observations on the effects of electrical stimuli on a 3D cellular construct.

First of all the effects of the electrical stimulus are dependent on the micro-environment (INFL or PHYS).

Second, we can generally observe that DC stimuli have a more impactful effect (in term of magnitude and extent of the functions involved) than AC stimuli. In particular, we can highlight a durable, moderate pro-inflammatory activity of DC1 in PHYS states, and a clear anti-inflammatory activity of DC5 on INFL states. Having observed an early and mild anti-inflammatory activity of DC1 on INFL, future experiments should assess the effect of intermediate voltages (3 V, 7 V for instance). Similarly, the effects of DC5 on PHYS deserve elucidation, given the activation of proliferative and *transitional* (EMT) functions. Furthermore, we addressed potential risks of DC treatment and found that a higher voltage (i.e., 7 V) is deleterious for the cells (cells died and hence no omics could be produced), highlighting the importance of establishing finer and more extended boundaries for each expected outcome, fundamental for a proper treatment protocol to avoid adverse consequences.

Regarding alternate current stimuli, while maintaining 5 V, AC10 appears overall to be associated with a mild proliferative activity (similar to DC1) in INFL accompanied by a reduction of *bile acids metabolism*, and opposite in PHYS where it reduces proliferation and activates *oxydative phosphorylation*. Higher frequency lead in INFL to reverse outcomes (reduced proliferation and fatty acid metabolism in INFL) and a new pattern in PHYS where *interferons*, *oxidative phosphorilation* and *fatty acid metabolism* are deactivated. Overall, at a voltage where in INFL we observe anti-inflammatory effects, AC modulates proliferation (enhanced and reduced at 10 vs. 100 Hz, respectively) with no direct impact on inflammation. In PHYS the effects appears to be opposed with a reduction of proliferative functions for AC10.

In conclusion, while voltage would deserve exploration for its role as a parameter to *dose* the therapy in inflamed states, frequency appears to lead to diverse and possibly non-linear effects including differential proliferative activities, strongly dependent of the surrounding micro-environment (*state*).

Interestingly, these results appear to indicate that the anti-inflammatory effect of an electrical stimulus can occur also in the absence of the mediation of the anti-inflammatory reflex. On the down-side, we must highlight the limitations of our pioneering work: our setting is extremely simple, and so is our inflammatory model, finally, the duration of our experiments is limited to Despite the simplicity of our setting, and the overall limited duration of our experiment (48 h). Nevertheless, the observation that electric stimuli can produce both an anti- and a pro- inflammatory action depending on the micro-environment is not novel in biological terms [35], and remains of high relevance towards medical translation. To the best of our knowledge, this work represents the first systematic effort to assess the effects of electrostimulation in a controlled, standardized and fully reproducible setting.Future investigation are warranted to obtain more insight into the mechanisms of action, including, for instance the analysis of other modalities (epigenomics) to obtain a more complete picture of the mechanisms at stake, finer time sampling, additional voltages and frequencies to clarify the parameters’ boundaries that elicit each type of output, mechanistic insight into the differences between direct and alternate current, to name a few. With these information at hand, more complex models could be tested, which would allow to expand the study on the temporal variable, beyond the 48 h constraints of our model, up until clinical trials.

## Figures and Tables

**Figure 1 ijms-25-09808-f001:**
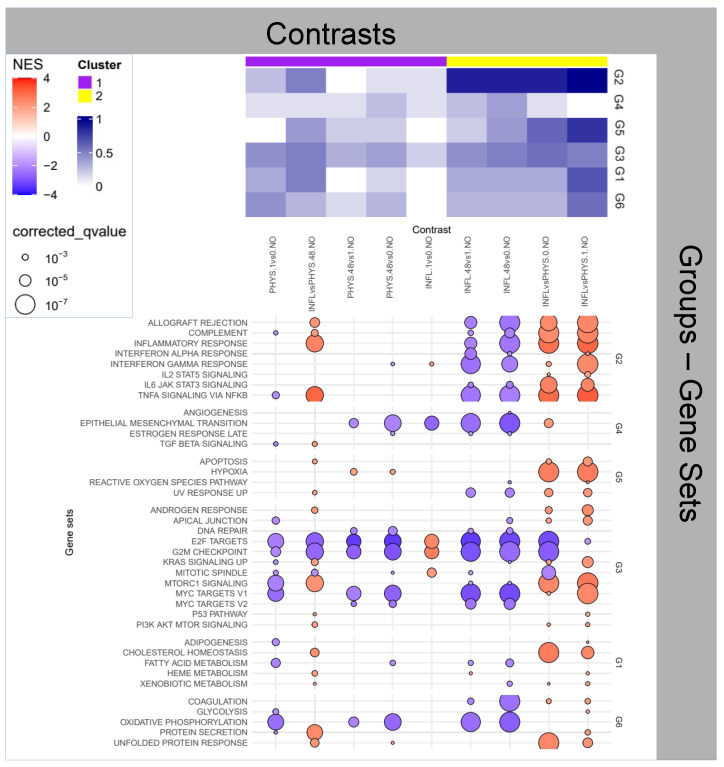
Enrichment Analysis for the impact of time on physiological samples and inflamed samples.

**Figure 2 ijms-25-09808-f002:**
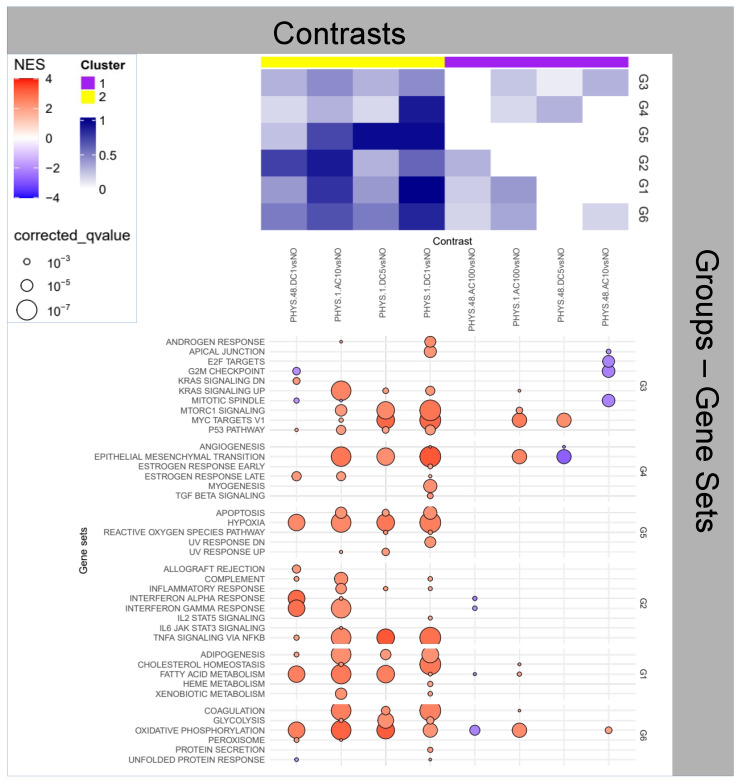
Enrichment Analysis for the impact of stimulus on physiological samples.

**Figure 3 ijms-25-09808-f003:**
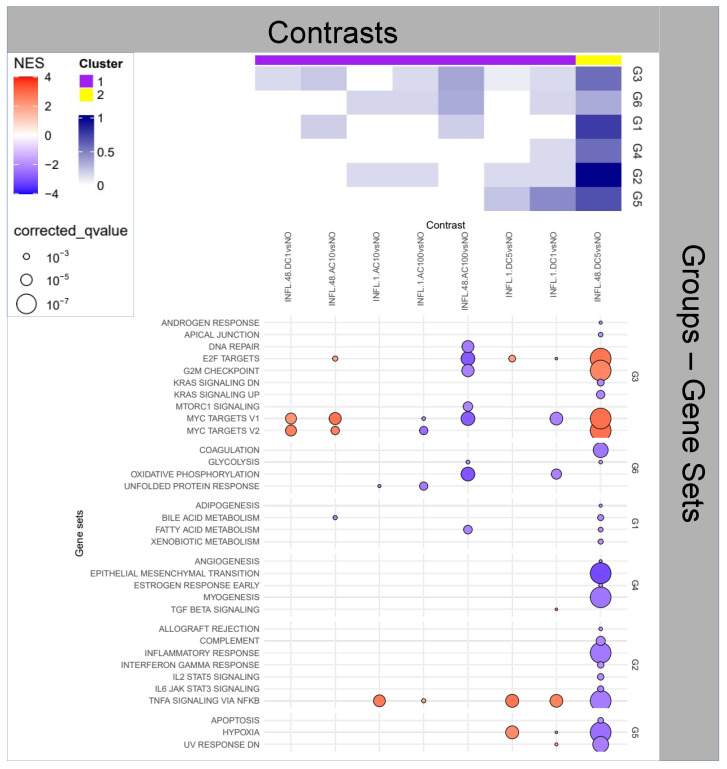
Enrichment Analysis for the impact of stimulus on inflamed samples.

**Figure 4 ijms-25-09808-f004:**
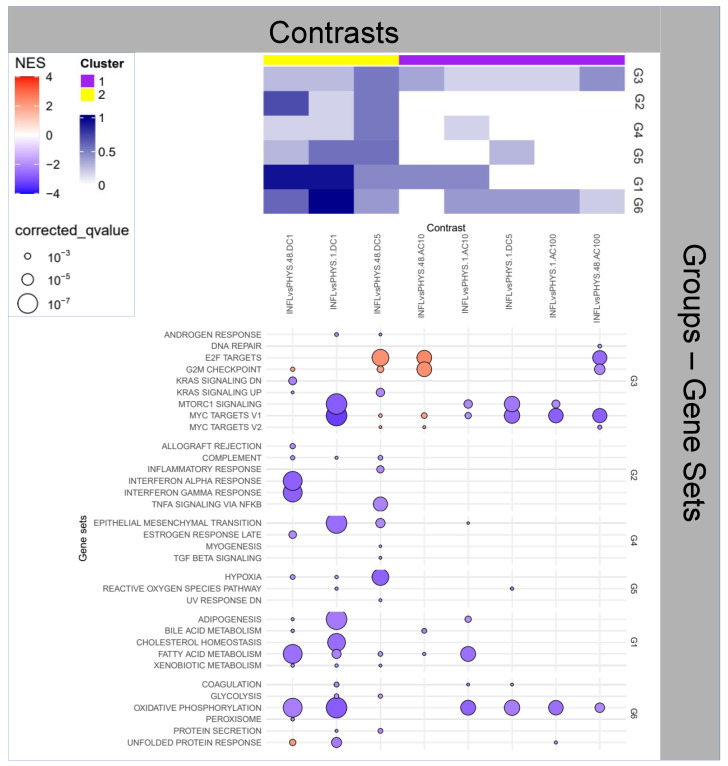
Enrichment Analysis for the impact of states on stimuli.

**Figure 5 ijms-25-09808-f005:**
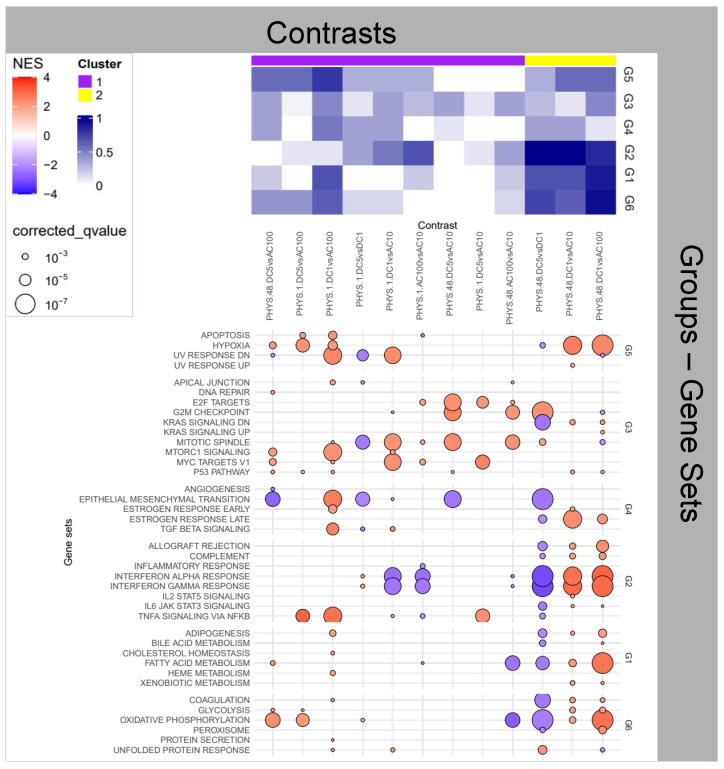
Enrichment Analysis for the impact of stimuli on PHYS.

**Figure 6 ijms-25-09808-f006:**
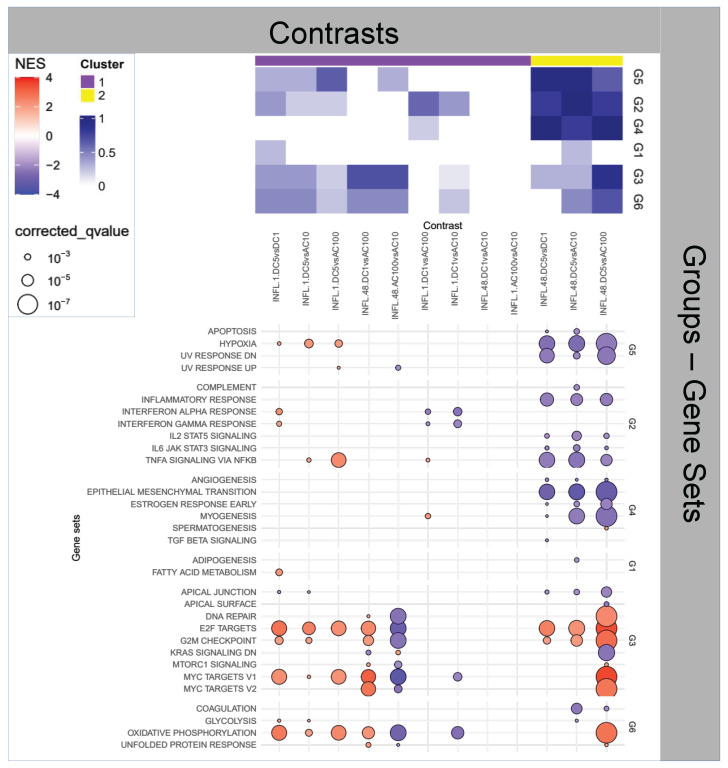
Enrichment Analysis for the impact of stimuli on INFL.

**Table 1 ijms-25-09808-t001:** Study design. The first four rows represent the main variables explored in the study: state (1st row); stimulus where NO stands for “no stimulus” (2nd row); further stimuli parameters (if any, 3rd row); sampling time (4th row). The 5th row lists on the numerator the number of samples whose quality actually enabled the production of transcriptomics, and the numerator the number of biological replicae.

INFL											PHYS										
NO			DC				AC				NO			DC				AC			
			1V		5V		10 Hz		100 Hz					1V		5V		10 Hz		100 Hz	
t0	t1	t48	t1	t48	t1	t48	t1	t48	t1	t48	t0	t1	t48	t1	t48	t1	t48	t1	t48	t1	t48
3/3	2/2	3/3	2/3	3/3	1/3	3/3	3/3	3/3	3/3	2/3	2/3	2/2	3/3	3/3	1/3	2/3	2/3	3/3	2/3	2/3	3/3

**Table 2 ijms-25-09808-t002:** Experimental questions, associated contrasts and summary results. Contrasts’ labels are constructed by the combination of the value of three variables (state, type of stimulus, time point) per conditions compared, where vs serves as separator of the differential condition.

Question	Contrasts	Effects
1. What is the impact of time?	PHYS.1vs0.NO, PHYS.48vs1.NO, PHYS.48vs0.NO, INFL.1vs0.NO, INFL.48vs1.NO, INFL.48vs0.NO, INFLvsPHYS.0.NO, INFLvsPHYS.1.NO, INFLvsPHYS.48.NO	Reduced proliferation increased hypoxia (PHYS) reduced inflammation (INFL)
2. What is the impact of stimuli on the physiological state?	PHYS.1.DC1vsNO, PHYS.1.DC5vsNO, PHYS.1.AC10vsNO, PHYS.1.AC100vsNO, PHYS.48.DC1vsNO, PHYS.48.DC5vsNO, PHYS.48.AC10vsNO, PHYS.48.AC100vsNO	DC1 inflammatory AC100 reduced interferons and energy production DC5 *transitional* activity AC10 reduced proliferation
3. What is the impact of stimuli on the inflamed state?	INFL.1.DC1vsNO, INFL.1.DC5vsNO, INFL.1.AC10vsNO, INFL.1.AC100vsNO, INFL.48.DC1vsNO, INFL.48.DC5vsNO, INFL.48.AC10vsNO, INFL.48.AC100vsNO	DC5 wound healing AC100 reduced proliferation AC10 & DC1 early inflammation followed by proliferation (no decrease in inflammation)
4. Given a stimulus, what is the (differential) impact of the state (INFL, PHYS)?	INFLvsPHYS.1.DC1, INFLvsPHYS.1.DC5, INFLvsPHYS.1.AC10, INFLvsPHYS.1.AC100, INFLvsPHYS.48.DC1, INFLvsPHYS.48.DC5, INFLvsPHYS.48.AC10, INFLvsPHYS.48.AC100	PHYS: AC pro-inflammatory with enhanced proliferation and frequency-dependent bile acids vs oxidative phosphorylation activity (10 Hz vs. 100 Hz) ACs mild anti-inflammatory DCs anti-inflammatory with enhanced proliferation (stronger in DC5 than AC100)
5. What is the differential impact of stimuli on PHYS?	PHYS.1.DC5vsDC1, PHYS.48.DC5vsDC1, PHYS.1.AC100vsAC10, PHYS.48.AC100vsAC10, PHYS.1.DC5vsAC10, PHYS.1.DC5vsAC100, PHYS.48.DC5vsAC10, PHYS.48.DC5vsAC100, PHYS.1.DC1vsAC10, PHYS.1.DC1vsAC100, PHYS.48.DC1vsAC10, PHYS.48.DC1vsAC100	DC5, AC10 no relevant impact on inflammation, contrasting proliferative activity (mildly down for AC10 and mildly up for DC5) AC100 mild anti-inflammatory DC1 mildly proinflammatory
6. What is the differential impact of stimuli on INFL?	INFL.1.DC5vsDC1, INFL.48.DC5vsDC1, INFL.1.AC100vsAC10, INFL.48.AC100vsAC10, INFL.1.DC5vsAC10, INFL.1.DC5vsAC100, INFL.48.DC5vsAC10, INFL.48.DC5vsAC100, INFL.1.DC1vsAC10, INFL.1.DC1vsAC100, INFL.48.DC1vsAC10, INFL.48.DC1vsAC100	DC1, AC10 do not show significant lasting (48 h) enrichment, indicating the same type of effect DC1 and AC10 more proliferative than AC100 AC100 lower proliferation and energy production (compared to baseline) DC5 anti-inflammatory and proliferative vs AC100

**Table 3 ijms-25-09808-t003:** Manually curated grouping of the Hallmark gene sets according to their biological functions.

Group	Hallmarks
1 Metabolism and homeostasis	Adipogenesis, Bile acid metabolism, Cholesterol homeostasis, Fatty acid metabolism, Heme metabolism, Xenobiotic metabolism
2 Immune Response and Inflammation	Allograft rejection, Complement, Inflammatory response, IFN-α response, IFN-γ response, IL-2 STAT5 signalling, IL-6 JAK STAT3 signalling, TNF-α signalling via NF-κB
3 Cellular Regulation and Proliferation	Androgen response, Apical junction, Apical surface, DNA repair, E2F targets, G2M checkpoint, KRAS signalling down, KRAS signalling up, mitotic spindle, mTORC1 signalling, Myc targets v1, Myc targets v2, NOTCH signalling, p53 pathway, Pancreas β-cells, PI3k Akt mTOR signalling
4 Cellular processes and Development	Angiogenesis, EMT, Estrogen response early, Estrogen response late, Myogenesis, Spermatogenesis, TGF-β signalling, Wnt-β catenin signalling
5 Signal Regulation and Stress Response	Apoptosis, Hypoxia, ROS pathway, UV response dn, UV response up
6 Specialized Functions	Coagulation, Glycolysis, Hedgehog signalling, Oxidative phosphorylation, Peroxisome, Protein Secretion, Unfolded Protein Response

## Data Availability

Transcriptomic data are available at Gene expression Omnibus with ID GSE254696.

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
