# Peer review of "Differential Anti-Inflammatory Effects of Electrostimulation in a Standardized Setting"

_ijms, 2024, doi:10.3390/ijms25189808_

Round 1

Reviewer 1 Report

Comments and Suggestions for Authors

With the current version, authors haven’t still revealed scope regarding to the Journal, i.e., molecules are the object of study. The scope includes Fundamental theoretical problems of broad interest in biology, chemistry and medicine; Breakthrough experimental technical progress of broad interest in biology, chemistry and medicine; Application of the theories and novel technologies to specific experimental studies and calculations. Therefore, the submission should be rejected. 

Comments on the Quality of English Language

Some parts of the English grammar in the manuscript need to be checked

Author Response

With the current version, authors haven’t still revealed scope regarding to the Journal, i.e., molecules are the object of study. The scope includes Fundamental theoretical problems of broad interest in biology, chemistry and medicine; Breakthrough experimental technical progress of broad interest in biology, chemistry and medicine; Application of the theories and novel technologies to specific experimental studies and calculations. Therefore, the submission should be rejected. 

  • We thank the reviewer for this frank comment, we have now clarified in the Abstract, Introduction and Conclusion the fundamental theoretical problem we wish to address in biology: if and what are the differential anti-inflammatory effects of electrical stimuli. To begin to address - as this represent a whole research area - this question, we designed an original 3D bioconstruct whose advantages lie mostly in the standardization and reproducibility of the experimental settings, a crucial lack in this research area. Limitations are clearly concerned with the fact that this is a model, which, like all models, can only capture portions of a real biological system. We highlighted therefore advantages and disadvantages in all the above-mentioned sections. To the best of our knowledge such a systematic research is a première, with the exception of a recent review (Katoh, 2003), collecting, however, disparate and diverse experimental setting available in the scientific literature, which does not enable direct comparisons among results, differently from our work.

Reviewer 2 Report

Comments and Suggestions for Authors

The study design is adequately made, but providing a more comprehensive elucidation of the reasons behind the selection of particular voltages and frequencies for the electrical stimulation would be advantageous. Using TNF-α as inflammation inducer is a conventional method, however, the manuscript should address the constraints of this model in accurately depicting intricate inflammatory circumstances observed in vivo. The analysis of the findings could be extended to explore the biological importance of the discovered pathways and their connection to the reported anti-inflammatory effects. Certain conclusions lack comprehensive discussion. Further exploration is needed to understand the consequences of the differential effects of direct and alternative current stimulation on proliferation and inflammation. The study emphasizes the capacity of 5V direct current stimuli to address inflammation but it fails to address the potential hazards or adverse consequences of this treatment, which is essential for its application in clinical settings. The conclusion summarizes the findings and it would benefit from further analysis of the implications of the study and of electrotherapy and prospective avenues for future research. The text would be improved by providing a more comprehensive analysis of the constraints, particularly addressing the potential influence of the brief duration of the experiment (48 hours) on the reported outcomes.

Comments on the Quality of English Language

The English quality of the manuscript requires minor revisions

Author Response

The study design is adequately made, but providing a more comprehensive elucidation of the reasons behind the selection of particular voltages and frequencies for the electrical stimulation would be advantageous.

  • In the absence of a clear rationale or de facto standard, our choices have been driven by numerous considerations taken from the limited available literature and from direct interaction with experts using electrostimulation for pain and inflammation control, based on empirical, experiential or traditional knowledge. All information considered is currently reported in the Supplementary Materials, given the lengthy list. To address the Reviewer request we have made this reference more explicit, we are of course willing to report this all in the Methods section shall this be considered more effective and not “distracting” from the main flow of the article.

    Using TNF-α as inflammation inducer is a conventional method, however, the manuscript should address the constraints of this model in accurately depicting intricate inflammatory circumstances observed in vivo.

  • We integrated the abstract by mentioning the use of TNF-a as conventional inflammation inducer and the introduction by pointing out potential limitations related to the capability to recapitulate complex in vivo inflammatory circumstances.

    The analysis of the findings could be extended to explore the biological importance of the discovered pathways and their connection to the reported anti-inflammatory effects. Certain conclusions lack comprehensive discussion. Further exploration is needed to understand the consequences of the differential effects of direct and alternative current stimulation on proliferation and inflammation.

  • We fully agree with the Reviewer, this is indeed our future work, for several years to come. The results of the present work are pioneering and fully exploratory. What we have been able to highlight here are the difference of effects, based on voltage and modality. This is just the tip of the iceberg, questions like the relation between voltage and effect, the causes behind the differential effects of alternate and direct current are as many questions that will require completely new experiments (and funding) to be addressed. This article offers our first steps into addressing the numerous questions that lie behind the relation between inflammation and electrostimulation, which represents a whole biomedical research area and which cannot be addressed in the frame of single publication (and possibly not in the time of a single researcher’s career). Therefore, although we share the same enthusiasm and curiosity that we have read across this comment, we are unable to address all the above points with a single publication. To avoid deception or overstatements we have added throughout the abstract, introduction and conclusion clarifying sentences stating both the novelty of our work (systemic and systematic exploration) and the limitations that you have also highlighted.

    The study emphasizes the capacity of 5V direct current stimuli to address inflammation but it fails to address the potential hazards or adverse consequences of this treatment, which is essential for its application in clinical settings.

  • We agree with the reviewer, we therefore added our experiments with superior voltages, and in particular we mentioned the tested 7V which was deleterious for the cells. Since we did not proceed with molecular data we originally had not added this point, which is now included in the conclusions), also it would be important to know whether the effects are linear with Voltage or other types of dependency exist. We also gave more emphasis in the conclusion on advantages and limitations.

    The conclusion summarizes the findings and it would benefit from further analysis of the implications of the study and of electrotherapy and prospective avenues for future research.

  • We have expanded the conclusion according to this recommendation.

    The text would be improved by providing a more comprehensive analysis of the constraints, particularly addressing the potential influence of the brief duration of the experiment (48 hours) on the reported outcomes.

  • We have expanded our considerations on the limitation of our experimental model and its short lifespan

Round 2

Reviewer 1 Report

Comments and Suggestions for Authors

The authors should revise the entire manuscript for English language accuracy, and cite more references suitably more.

Comments on the Quality of English Language

The authors should revise the entire manuscript for English language accuracy.

Reviewer 2 Report

Comments and Suggestions for Authors

I consider the article proper to be published in the present form, the authors solved all my inquiries